# Three Molecular Modification Strategies to Improve the Thermostability of Xylanase XynA from *Streptomyces rameus* L2001

**DOI:** 10.3390/foods12040879

**Published:** 2023-02-18

**Authors:** Weijia Zhu, Liqin Qin, Youqiang Xu, Hongyun Lu, Qiuhua Wu, Weiwei Li, Chengnan Zhang, Xiuting Li

**Affiliations:** 1School of Food and Health, Beijing Technology and Business University (BTBU), Beijing 100048, China; 2Key Laboratory of Brewing Microbiome and Enzymatic Molecular Engineering, China General Chamber of Commerce, Beijing 100048, China; 3Key Laboratory of Brewing Molecular Engineering of China Light Industry, Beijing Technology and Business University, Beijing 100048, China

**Keywords:** GH11 xylanase, thermostability, surface entropy, disulfide bond, molecular cyclization

## Abstract

Glycoside hydrolase family 11 (GH11) xylanases are the preferred candidates for the production of functional oligosaccharides. However, the low thermostability of natural GH11 xylanases limits their industrial applications. In this study, we investigated the following three strategies to modify the thermostability of xylanase XynA from *Streptomyces rameus* L2001 mutation to reduce surface entropy, intramolecular disulfide bond construction, and molecular cyclization. Changes in the thermostability of XynA mutants were analyzed using molecular simulations. All mutants showed improved thermostability and catalytic efficiency compared with XynA, except for molecular cyclization. The residual activities of high-entropy amino acid-replacement mutants Q24A and K104A increased from 18.70% to more than 41.23% when kept at 65 °C for 30 min. The catalytic efficiencies of Q24A and K143A increased to 129.99 and 92.26 mL/s/mg, respectively, compared with XynA (62.97 mL/s/mg) when using beechwood xylan as the substrate. The mutant enzyme with disulfide bonds formed between Val3 and Thr30 increased the *t*_1/2_^60 °C^ by 13.33-fold and the catalytic efficiency by 1.80-fold compared with the wild-type XynA. The high thermostabilities and hydrolytic activities of XynA mutants will be useful for enzymatic production of functional xylo-oligosaccharides.

## 1. Introduction

Chronic diseases are a major health problem. Intestinal microbes, which form an important microbial barrier to maintain human health, play an important role in preventing chronic disease [1,2]. Fermented foods, short-chain fatty acids, and probiotics can be used to regulate intestinal microbes [3,4,5,6]. Xylo-oligosaccharides (XOS) are typical probiotics that are effective for regulating intestinal microbes and preventing chronic disease [7]. The main methods for the industrial production of XOS are natural hydrolysis, acid hydrolysis, and enzymatic hydrolysis. Enzymatic production of XOS is environmentally friendly and highly efficient [8,9], and has good industrial prospects for the preparation of functional XOS. Glycoside hydrolase family 11 (GH11) xylanases have strong substrate specificity and high catalytic efficiency for XOS production [10,11]. However, most natural GH11 xylanases have poor stability and are not suitable for use at the high temperatures encountered in industrial production [11]. Enzymatic engineering can be used to modify natural xylanases so that they are stable under high temperature conditions, and retain good substrate specificity and high catalytic efficiency. Modification will be important for the industrial production of functional XOS [10,12]. However, many complex factors affect the thermostability of GH11 xylanases, and mutant enzymes designed using sequence preferences and structural rigidity modification strategies often fail to generate enzymes with good stability and high catalytic activity [13]. Consequently, thermostable modification of GH11 xylanase remains challenging. Rational design using the enzyme structure and function is a feasible option to enhance the thermostability of GH11 xylanase [14,15,16].

The thermostability of an enzyme is often influenced by specific amino acid residues. Perl et al. compared cold shock proteins derived from thermophilic *Bacillus caldolyticus* and mesophilic *Bacillus subtilis*, and found that the two proteins differed in only 12 residues but displayed a considerable difference in stability. Site-directed mutagenesis of these 12 residues revealed that the difference in thermostability was accounted for by the substitution of only two residues (Arg3 and Leu66) on the surface of the molecule [17]. Therefore, the substitution of surface residues may provide a simple yet powerful approach for increasing the thermostability of a protein. For small proteins that exhibit high homology and significant differences in thermostability, substitution of less than 20.00% of the amino acids is sufficient to modify the enzyme stability [18]. This result demonstrates that key amino acid residues are crucial for rationally designing an enzyme to improve the thermostability [14]. However, the rational design of xylanases for thermostability is challenging because there have been few predictions of mutation sites, and their mechanism of action remains unclear [19,20].

The modification of xylanases to improve thermostability generally involves increasing the thermodynamic stability of the enzyme by increasing the free energy differences between the unfolded and folded states of the enzyme, and reducing the rate of de-folding by increasing the free energy differences between the folded and unfolded transition states [21]. To achieve these two goals, many rational and semi-rational design methods have been applied to modify enzyme thermostability [22]. For example, the conformational entropy of the unfolded state is reduced by introducing disulfide bonds [23,24] or by introducing proline to make flexible sites rigid [20,21]. The stability of the internal regions of the enzyme is enhanced by the introduction of hydrogen bonding [25,26,27], hydrophobic interactions [28,29,30], or electrostatic interactions [31,32,33], or by altering the solubility of specific regions of the enzyme [31]. Sutthibutpong investigated the effects of hydrophobic interactions, hydrogen bonding, and disulfide bonds on the thermostability of xylanase xyn11A, and found that the disulfide bonds had the highest contribution to thermostability [34]. Paës et al. increased the half-life of the enzyme from 18 to 180 min at 70 °C by constructing a disulfide bond within Tx-xyl, which is a GH11 xylanase from *Thermobacillus xylanilyticus* [35]. Moon et al. mutated charged residues on the protein surface, including lysine and glutamic acid, to non-polar amino acids, such as alanine, which lowered the protein surface entropy and enhanced the protein stability [36]. Wang et al. showed that mutants (S142A/D217V/Q239F/S250Y) of *Rhizopus chinensis*-derived lipase r27RCL had lower conformational entropy, a reduced solvent-accessible surface area, and increased structural rigidity [37]. These studies suggest that the introduction of disulfide bonds and the alteration of high-entropy amino acids located on the enzyme surface affect the thermostability of the enzyme. The N-/C-terminals of enzymes with linear conformations usually unfold first during thermal denaturation, which has a large impact on the stability of the enzyme. Zhou et al. used Spy Tag/Spy Catcher to cyclize GH10 family xylanase LXY, resulting in 2.76-, 1.70- and 9.58-fold increases in thermostability at 60 °C, 70 °C, and 80 °C, respectively, when compared with the linear enzyme [38]. The cyclization of linear proteins, by adding peptide chains to link protein ends, can effectively anchor the easily unfolded regions of the protein, resulting in enhanced thermostability [36]. The above studies show that disulfide bonding, surface entropy, and molecular cyclization are effective methods for enhancing enzyme thermostability.

In our previous studies, we found that GH11 xylanase XynA from *Streptomyces rameus* L2001 had high specific activity for beechwood xylan (1358.80 U/mg), and degraded it to yield mainly X2 and X3, indicating its potential application for the production of functional XOS [39,40]. However, the enzyme activity decreased to 18.70% when the temperature was held at 65 °C for 30 min. In this study, we aimed to improve the thermostability of XynA using three different methods. The results provide a reference for the modification of xylanases to improve thermostability and will promote the industrial application of XynA.

## 2. Materials and Methods

### 2.1. Gene, Strains, Media and Substrates

Xylanase gene *XynA* from *Streptomyces rameus* L2001 was used, and the nucleotide and amino acid sequences of *XynA* were deposited in the National Center for Biotechnology Information (NCBI) with accession numbers KC011007.1 and AFW21197.1, respectively. *Escherichia coli* (*E. coli*) DH5α and *E. coli* BL21 (DE3), pMD18-T and pET-28a (+), plasmid extraction kit, DNA gel extraction kit, isopropyl-β-D-thiogalactopyranoside (IPTG), kanamycin (KAN^+^), and ampicillin (AMP) were purchased from Takara (Tokyo, Japan). Primer STAR Max polymerase and restriction endonucleases were obtained from NEB Inc. (Ipswich, MA, USA). Bovine serum albumin was obtained from Solarbio Co. (Beijing, China). Protein Thermal Shift™ Dye Kit was obtained from Thermo Fischer Scientific Inc. (Rockford, IL, USA). Birchwood and oat-spelt xylans were purchased from Sigma (St. Louis, MO, USA). Beechwood xylan was purchased from Megazyme (Wicklow, Ireland). *E. coli* BL21 transformants were cultivated in Luria-broth (LB) medium (10 g/L peptone, 5 g/L yeast extract and 10 g/L NaCl) for enzyme expression.

### 2.2. Construction of Mutant Xylanase

The mutants Q24A, K104A, K143A, XynA-VT, XynA-NT, and XynA-SN were constructed using the polymerase chain reaction (PCR) method with the plasmid pET28a-XynA (WT) as the template [41,42]. The sequences of the primers are listed in Appendix A. The PCR reaction conditions were 98 °C for 30 s; 32 cycles of 98 °C for 10 s, 58 °C for 15 s, 72 °C for 2 min; and an elongation step at 72 °C for 5 min. The PCR products were incubated with restriction enzyme *DpnI* to digest the original DNA template and were then separately transformed into *E. coli* BL21.

### 2.3. Expression and Purification of Xylanases

The correct recombinant plasmids transformants were inoculated into LB medium containing 40 μg/mL KAN^+^, cultured at 37 °C and 200 rpm. Thereafter, IPTG was added to a final concentration of 0.50 mM and the cultures were kept at 20°C for 14.00–16.00 h to induce xylanase expression. The cells were collected by centrifugation at 9000× *g* for 5 min at 4 °C, resuspended in 50.00 mM phosphate buffer at pH 6.0, and ultrasonically broken at the cell wall. The crude enzyme solution was collected by centrifugation at 9000× *g* at 4 °C for 30 min [40].

The recombinant proteins were purified using a Ni Sepharose High Performance (HP) column (1 × 10 cm) with 50.00 mM phosphate buffer (pH 7.5) consisting of 300.00 mM NaCl and 0.00–500.00 mM imidazole on an ÄKTA fast protein liquid chromatography (FPLC) purification system (GE Healthcare, Uppsala, Sweden) at a flow rate of 1.00 mL/min [40,43,44]. The Bradford method was used to measure the enzyme concentration with bovine serum albumin as the standard [45]. The purity of the enzymes was determined by sodium dodecyl sulfate polyacrylamide gel electrophoresis (SDS-PAGE) [46]. The protein concentrations were determined by a bicinchoninic acid assay (BCA) protein assay kit (Thermo Fischer Scientific Inc., Rockford, IL, USA) using bovine serum albumin (BSA) as the standard [44].

### 2.4. Xylanase Activity Assay

The xylanase activity was analyzed according to the previous report [47]. The reaction mixture containing 0.10 mL of enzyme solution and 0.90 mL of 1.0% (*w*/*v*) beechwood xylan was incubated in 50.00 mM citrate buffer (pH 6.0) at 50 °C for 10 min. The amount of reducing sugar was evaluated by the 3,5-dinitrosalicylic acid (DNS) method, using xylose (X1) as the standard. One unit (U) of xylanase activity was defined as the amount of enzyme releasing 1.00 μM of X1 equivalent reducing sugars from the substrate per minute [48].

### 2.5. Biochemical Characterization of Xylanase

The optimal pH of xylanase was measured at 37 °C using 50.00 mM buffer with pH values ranging from 2.0 to 12.0, including Glycine-HCl (Gly-HCl) buffer (pH 2.0–3.0), citrate buffer (pH 3.0–6.5), phosphate buffer (pH 6.5–8.0), barbital sodium buffer (pH 8.0–9.5), Glycine-NaOH (Gly-NaOH) buffer (pH 9.5–10.5), 3-(cyclohexylamino)-1-propanesulfonic acid (CAPS) buffer (pH 10.5–11.0), and NaH_2_PO_4_-NaOH buffer (pH 11.0–12.0). The pH stability of xylanase was determined by incubating the enzyme in buffers with pH ranging from 2.0 to 12.0 for 30 min at 37 °C, and then measuring the residual enzyme activity. The optimal temperature of xylanase was detected for the enzyme activity in 50 mM citrate buffer at temperatures ranging from 30 °C to 80 °C. The thermostability of xylanase was determined by incubating the enzyme at temperatures ranging from 30 °C to 70 °C for 30 min in 50.00 mM citrate buffer.

The protein concentration of the pure enzyme solution was diluted to 0.50 mg/mL using citrate buffer at pH 6.0, and the diluted enzyme solutions of the mutant enzyme and XynA were kept at 60 °C for 0, 15, 30, 60, 90, 120 and 180 min, respectively, and cooled rapidly in an ice water bath. The activity of untreated xylanase was defined as 100%, and the residual enzyme activity was calculated using the formula y = Ae^−kt^ (A is the initial enzyme activity, t is time, k is the decay constant). The half-life *t*_1/2_ is equal to ln2/k. The half-lives of mutants and XynA at 60 °C were calculated separately according to the formula [49]. The Protein Thermal Shift ™ dye kit was used to determine the thermal denaturation temperature of mutants and XynA [50]. Quantitative real-time PCR was performed using the CFX Touch 96-well system (Bio-Rad, California, CA, USA). Samples were heated on a 0.05 °C/s gradient from 4 to 95 °C and protein unfolding at each temperature was monitored by measurement of fluorescence at 580/623 nm (excitation/emission) [51].

Substrate specificity of the enzymes was investigated using the standard assay procedure with one of the following substrates (1%, *w*/*v*), beechwood xylan, birchwood xylan, or oat-spelt xylan. The liberated reducing sugars’ concentrations were determined by the DNS method, as described in Section 2.4. The kinetic parameters of xylanase were measured with different concentrations (2.50–30.00 mg/mL) of beechwood xylan in optimal reaction conditions. The values of *K*_m_ and *V*_max_ of the enzyme were calculated according to the Michaelis-Menten equation using GraphPad Prism software [40,52].

### 2.6. Hydrolytic Characteristics of XynA Mutants

The hydrolytic characteristics of xylanases were evaluated as previously described [48]. In brief, 5.00 U/mL of xylanase was mixed with 10.00 mg/mL of different substrates (beechwood xylan, birchwood xylan, or oat-spelt xylan) and then incubated in 50.00 mM citrate buffer (pH 5.0) at 50 °C. The hydrolysis products were analyzed quantitatively by high-performance liquid chromatography (HPLC). The 10.00 μL sample was injected into a KS-802 column (8 mm × 300 mm) with a differential refractive index detector (RID). The temperatures of the chromatographic column and RID were 65 °C and 30 °C, respectively. X1, X2, X3, X4, and X5 were used as standards. All experiments were performed in triplicate.

### 2.7. Sequence and Structural Analysis

Amino acid sequences of xylanase were aligned using DNAMAN. NCBI (National Center for Biotechnology Information; http://www.ncbi.nlm.nih.gov/, assessed on 7 November 2022) and PDB (Protein Data Bank; http://www. rcsb.org/, assessed on 7 November 2022) were used to download sequence and structure information of GH11 xylanases. Homology modeling was performed using the GH11 xylanase XlnB2 (PDB ID 5ej3) and Xyn11A (PDB ID 7eo6) as the templates by SWISS-MODEL (https://beta.swissmodel.expasy.org/, assessed on 5 October 2022). The structured models were visualized by Pymol software. Three-dimensional structural analysis of the recombinant xylanase XynA before and after the mutation was performed using Pymol. Surface entropy reduction (SER) (https://services.mbi.ucla.edu/SER/, assessed on 29 October 2022) was used for surface entropy prediction [53]. The interaction force was analyzed to compare the differences between native and mutant xylanases. Hydrophobic and hydrophilic analyses of xylanases were calculated by Protein GRAVY (https://www.bioinformatics.org/sms2/protein_gravy.html, assessed on 20 November 2022). The hydrophilicity value of each amino acid of the enzyme could be calculated according to the algorithm and named as the grand average of hydropathy value (GRAVY). Negative GRAVY values signify hydrophilicity, and positive values indicate hydrophobicity; the lower value indicates that the enzyme is more hydrophobic [54]. Molecular docking analysis of xylanase and substrate xylohexaose was conducted by using AutoDock 1.5.6, and the hydrogen bonds between the subsites of xylanase and xylohexaose were analyzed using Discovery Studio 2020. Molecular dynamics (MD) simulations were performed to further assess the global stability of wild-type XynA and mutants using GROMACS 4.5.4 at 70 °C for 100 ns. The charm36m force field was used [55], and trajectory data were saved every 2 fs. Root mean square deviation (RMSD), root mean square fluctuation (RMSF), and dynamical cross-correlation matrix (DCCM) were analyzed using Bio3D [56]. The RMSD was calculated for all heavy atoms of proteins, and the RMSF was calculated for the Cα of protein. The DCCM was used to understand the correlation between specific atoms of each amino acid in a protein, providing information about the correlated motion of the protein on a large scale [57].

### 2.8. Statistical Analysis

All experiments were conducted in triplicate and data are presented as means ± standard deviation (SD). Statistical analysis was performed using one-way ANOVAs and Duncan’s multiple range tests (SPSS 26.0). Results were considered statistically significant at the 95 % confidence level (*p* < 0.05) [52].

## 3. Results and Discussion

Enzyme function requires an active folded form, but this form is unstable because it is easily unfolded into a flexible, unstructured form. Protein structural stability depends on the enthalpy contribution of covalently bound amino acids on the protein backbone, and pairwise non-covalent interactions between atoms (van der Waals, electrostatic, and solvent effects) [58]. Therefore, according to the structural characteristics of an enzyme, stabilizing the regions of the enzyme structure that are prone to de-folding can help to enhance the thermostability of the enzyme.

The XynA structure resembles a right-handed half-grip, and consists of two parallel β-sheets and an α-helix (Figure 1c). The catalytic active centers Glu86 and Glu177 are situated in the cleft created by the twisted portion, which is where catalytic activity occurs. Alterations close to the active site during the modification of XynA should be avoided to avoid interfering with the enzyme’s capacity to perform its binding and catalytic functions. The loop structure, which is found on the surface of the protein molecule and has a high degree of flexibility in dynamic changes of the protein structure, connects the β-sheets in the structure of XynA. This structure is unfavorable for stability. Molecular dynamics analysis has shown that the N- and C-terminal, α-helix regions are more likely to cause disruption of the spatial structure than the catalytic region under thermal treatment, which will trigger the de-folding of GH11 xylanase [59]. The following three methods have been used to modify the thermostability of XynA. These include predicting amino acids with a high entropy on the protein surface to target mutations, constructing disulfide bonds in the easily unfolded region of the protein, and introducing heterologous peptides to link linear protein terminal cyclase molecules. Exploring the effect of improving the thermostability of the mutant enzyme by measuring its enzymatic properties using simulations. 

### 3.1. Modification of Thermostability Based on the Surface Entropy

#### 3.1.1. Production of Surface Entropy Mutants

Protein unfolding promotes an increase in the conformational degrees of freedom of the amino acid backbone. Stable amino acid substitutions have a positive effect on stabilizing the conformational stability of protein folding [22]. Surface entropy reduction combines conformational entropy mapping, secondary structure prediction, and sequence conservation for predicting highly flexible hydrophilic residues and replacing residues with high conformational entropy, such as Lys and Gln, with amino acids without side chain-conformational entropy or with low conformational entropy, such as Ala, Tyr, and Thr. This reduces the conformational entropy of the protein [60]. The predicted sites are usually located on the protein surface, and are poorly conserved evolutionarily and far from the catalytic center of the enzyme. A comprehensive analysis of XynA sequence conservation, solvent accessibility, and secondary structure features was performed to compare the predicted surface entropy reduction fractions capable of reducing conformational entropy [61]. Three high-entropy residues, Q24, K104, and K143 were located in the non-catalytic region on the protein surface (Figure 1a, b). Q24 and K143 were located in the loop region of the protein secondary structure, and K104 was located in the β-sheets near the α-helix (Figure 1c). SDS-PAGE analysis showed that XynA migrated as a band and the molecular weight was approximately 20.80 kDa (Figure 1d).

#### 3.1.2. Characterization of Surface Entropy Mutants

Seven different buffers with pH values ranging from 2.0 to 12.0 were chosen to examine the effect of pH on the enzymatic activities of the three mutants (Figure 2). The optimum pH for all three mutants was 5.0, which was lower than that for XynA (5.5) [40]. Compared with Ala, both Lys and Gln have amino groups in their side chains, and Zhou et al. found that after mutation of these two sites to Ala, the enzyme’s surface charge distribution changed, and the protein’s optimum pH moved to become more acidic [62]. In the present study, all three mutants were stable over a wider pH range than XynA. Residual enzyme activity was maintained above 81.27% after 30 min incubation at pH 2.0–8.5 for Q24A and at pH 2.0–11.0 for K104A and K143A. The residual activity of XynA fell below 77.10% after 30 min of incubation at pH 2.0 and 10.0 (Appendix A). 

The optimum temperature for Q24A and K104A was 65 °C, and the optimum temperature for K143A was 60–65 °C (Figure 3a). The optimum temperatures of all three mutant enzymes were higher than that for XynA (60 °C). The residual activities of Q24A and K104A were all above 41.23% when the temperature was held at 65 °C for 30 min, which was an increase compared with XynA (18.70%). However, the thermostability of K143A decreased compared with XynA. After incubation at 60 °C for 30 min, the residual activity of K143A was only maintained at approximately 58.37%, which was 23.14% lower than that of XynA (Figure 3a). The half-lives of Q24A and K104A were 76.17 and 88.87 min, respectively, and were similar to that of XynA (86.64 min). The half-life of K143A (69.31 min) decreased by 17.33% compared with XynA. This indicated that the thermostability of K143A at 60 °C was lower than that of XynA. The thermostability of recombinant xylanase was evaluated at the protein thermodynamic level, and the thermal denaturation temperatures (*T*_m_) of Q24A, K104A, K143A and XynA were determined using Protein Thermal Shift ™. The *T*_m_ for Q24A, K104A, K143A, and XynA were all approximately 73.50 °C. The enzyme structure was relatively stable, which was not reflected in the thermodynamic parameters (Figure 4) (Table 1).

Results for the interactions in XynA and the mutant xylanase were compared. Amino acid alterations affected the interaction of amino acids, and this influenced the stability of the enzyme (Figure 5). Compared with XynA, K143A had reduced hydrogen bonding interactions with surrounding amino acids, and mutation resulted in the original hydrogen bonding interactions of K143 with W94 and G95 disappearing. These interactions were important for maintaining the structural stability of the protein [63]. The reduction in hydrogen bonding resulted in a decrease in the thermostability of K143A compared with XynA. Vieira et al. used molecular dynamics simulations for two structurally similar GH11 xylanases: thermophilic xylanase BCX from *Bacillus circulans*, and thermophilic xylanase TLX from *Thermomyces lanuginosus* [27]. Their results demonstrated that intramolecular hydrogen bonding was a key factor in maintaining the rigidity of the protein backbone at high temperatures. Hydrogen bonding at the Q24A and K104A mutation sites was unchanged, and it was speculated that the enhanced thermostability was a result of reduced conformational entropy and enhanced structural rigidity. The protein structure was stabilized by replacing amino acids with high entropy values, which reduced the protein conformational entropy. The software-assisted prediction of flexible sites with high entropy and thermal fluctuations and their rigidification by molecular modification were effective methods to improve the thermostability of enzymes [64]. 

Additionally, the GRAVY values of Q24A, K104A, and K143A were −0.55, and were lower than the GRAVY value of XynA (−0.58). This difference indicated that when the high-entropy amino acid was mutated to low-entropy Ala, the local hydrophobicity of the protein surface increased (Figure 6). At the same time, interaction of the amino acid side chain with the solvent at this position was reduced, which was beneficial to the enzyme stability. Pace et al. measured conformational stability changes (∆*G*) for four proteins with the addition of hydrophobic interactions and found that hydrophobic interactions were the dominant determining factor of protein stability at approximately 60.00% [65]. Protein stability is enhanced by hydrophobic interactions because the hydrophobic environment provides 1.30 ± 0.50 kcal/mol of energy to stabilize the protein for each additional -CH_2_- group [66]. An increase in hydrophobic interactions decreases the rate of protein unfolding in a thermal environment [67]. Xing et al. combined qualitative evolution and rational design to obtain a mutant of GH11 xylanase, Xyn376, with a half-life of 720.00 times that of the wild enzyme at 70 °C [68]. Structural analysis revealed that hydrogen bonding and hydrophobic interactions were mainly responsible for the enhanced thermostability. Although the local hydrophobicity of K143A was enhanced, hydrogen bonding interactions with the surrounding amino acids decreased. It was speculated that the energy provided by hydrogen bonding at this position was greater than that provided by the hydrophobic interaction, and this reduced the thermostability. 

### 3.2. Thermostability of Disulfide Bonds Mutations of XynA

#### 3.2.1. Construction of Disulfide Bonds Mutants

The disulfide bond is a covalent bond that can connect the enzyme’s secondary structure, reduce its fluctuation and unfolding, and reduce the conformational entropy of the protein to improve the structural stability of the protein, which is important for the thermostability of enzymes [23]. Tang et al. introduced a disulfide bond F14C–Q52C between the two β-sheets (A2 and A3) of GH11 xylanase Xyn2, and found that the half-life of the mutant at 60 °C (9.60 min) increased to twice that of the original enzyme (3.90 min) [69]. Teng et al. introduced a disulfide bond, T2C–T29C, between the N-terminal loop region of xylanase PjxA and β-sheet A2, and found that the optimum temperature of mutant PjxA-DB was 65 °C, which was 15 °C higher than that of the original enzyme [52]. Sequence alignment of XynA with modified enzymes containing disulfide bonds with better thermostability will help to identify potential sites for constructing disulfide bonds, and to design corresponding cysteine mutations to improve the thermostability of the enzyme. By comparing the sequences of XynA with the modified enzymes PTxA-DB [70], XynZFTA [71], and xyn11A-SS [34] (Figure 7a), potential sites for constructing disulfide bonds were found at the N-terminal, C-terminal, and between the α-helix and β-sheet of XynA, V3C−T30C, N36C−T188C, and S109C−N153C, respectively. Three mutant enzymes, XynA-VT, XynA-NT, and XynA-SN were designed with disulfide bonds (Figure 7b).

SDS-PAGE analysis of the purified mutants is shown in Figure 7c. The presence of disulfide bonds was verified using the method of Wakarchuk et al. [72]. Because dithiothreitol (DTT) is highly reductive and can reduce disulfide bonds in proteins, a mutant enzyme with reduced disulfide bonds binds more SDS and migrates more slowly during SDS-PAGE. Therefore, the pure enzyme solution was treated with different concentrations of DTT to verify the formation of disulfide bonds. The pure enzyme solution treated with 10.00 mM DTT showed a thicker protein band at approximately 20.00 kDa, which was close to the predicted value of 20.80 kDa. This result indicated that expression of XynA-VT, XynA-NT, XynA-SN, and XynA-SN mutants with disulfide bonds and mutant enzymes was not reduced. XynA-SN was successfully expressed and purified. The purified enzyme solution treated with 2.50 mM DTT formed two bands because of incomplete disulfide bond disruption, which indicated that disulfide bond formation was successful.

#### 3.2.2. Characterization of Mutants with Disulfide Bonds

The optimum pH for both XynA-VT and XynA-SN was 5.5, which was the same as that of XynA, the optimum pH of XynA-NT was 5.0, which was lower than that of XynA (Figure 8). Mutation of Asn36 to Cys36 led to a significant shift in the optimum pH of the enzyme towards an acidic value, which was consistent with a previous study [62]. The pH stability results showed that the residual activities of XynA-VT and XynA-NT were maintained above 82.26% in the pH range of 2.0–9.5, and the pH stability of XynA-SN was maintained above 79.31% in the pH range of 2.5–7.5. However, when the pH was above 8.5, the residual activity was only 66.29% (Appendix A). These results showed that the pH stability of XynA-SN decreased under alkaline conditions. 

The optimum temperatures of all three mutants with introduced disulfide bonds were higher than that of XynA (Figure 3b). The optimum temperatures was 70 °C for both XynA-VT and XynA-NT, which was 10 °C higher than that of XynA. The optimum temperature of XynA-SN was 65 °C, which was 5 °C higher than that of XynA. Many studies have shown that the construction of disulfide bonds in an enzyme effectively reduces the conformational entropy, allowing it to form an optimal conformation in a higher temperature environment. This maximizes the activity and increases the optimum reaction temperature of the enzyme [70,71,73]. The mutants were more thermostable after the construction of additional disulfide bonds, and all of them had residual activity above 92.80% when kept at 65 °C for 30 min. XynA showed only 18.70% residual enzyme activity under the same conditions. XynA-VT could maintain its residual activity above 82.45% when kept at 70 °C for 30 min, which showed that XynA-VT had higher thermostability compared with all mutants. The residual activity of XynA-SN fell to less than 61.26% when maintained at 30–40 °C for 30 min, but other enzymes were able to maintain more than 79.68% residual activity, which indicated that the thermostability of XynA-SN decreased at lower temperatures. The construction of disulfide bonds at the α-helix of XynA-SN generated a rigid enzyme structure and reduced the thermodynamic motion of the enzyme at low temperatures, leading to lower activity. This situation has also been found in hyper-thermophilic enzymes [74]. He et al. increased the stability of *Trichoderma reesei*-derived xylanase Xyn2 at 60–80 °C, while decreasing it at 50–55 °C by introducing a disulfide bond at the N-terminal [75]. The half-lives of the enzymes are shown in Table 1. The half-lives of the mutants at 60 °C were significantly higher than that of XynA (86.60 min). The half-life of XynA-VT was 1155.20 min at 60 °C (13.33 times than that of XynA), and the half-life of XynA-NT was 693.15 min at 60 °C (8.00 times than that of XynA). The half-life of XynA-VT was 92.40 min at 70 °C, which was higher than that of XynA at 60 °C (86.60 min). The half-life of XynA-VT was 92.40 min at 70 °C, which was also higher than that of XynA-NT (35.00 min at 70 °C) and XynA-SN (3.50 min at 70 °C). Therefore, the construction of a disulfide bond made the mutants more thermostable at high temperatures compared with XynA.

Figure 9 shows changes in the enzyme structures with construction of the disulfide bonds. The formation of disulfide bonds between Cys3 and Cys30 of XynA-VT did not disrupt the original hydrogen bonding interaction between Thr30 and Ser38, and the bonds and interactions stabilized the N-terminal structure. The formation of disulfide bonds between Cys36 and Cys188 disrupted the hydrogen bonding interactions between Thr188, Asn186, and Ile187. It has been confirmed that disulfide bonds, as a covalent interaction, provide much higher energy (12.55 kJ/mol) than non-covalent interactions such as hydrogen bonds [22]. Disulfide bonds played a crucial role in the thermostability of xylanases, even though XynA-NT reduced the hydrogen bonding interaction during the formation of disulfide bonds. However, its contribution to the thermostability was less than that of XynA-VT because of the reduction in hydrogen bonding. A similar phenomenon occurred in the mutant XynA-SN, where the formation of disulfide bonds between S109C and N153C caused the hydrogen bonds between Ser109 and Asp110 to disappear, but still improved the thermostability of the enzyme. The GRAVY value of the mutant XynA-VT with the disulfide bond introduced at the N-terminal was −0.57, and this was not significantly different from that of XynA (−0.58). The GRAVY value of XynA-NT was −0.53, which indicated that the enzyme had improved hydrophobicity (Figure 6). 

### 3.3. Thermostability of XynA with Heteropeptide-Mediated Cyclization

#### 3.3.1. Construction of XynA Cyclization Mutants

The distance between the C- and N-terminals of xylanase XynA is 22.80 Å, according to a 3D structure model of XynA [40]. This information was used as the structural basis for cyclization. XynA cyclized using the addition of SpyRing-tagged peptides. The SpyTag tag was added in front of the N-terminal sequence of XynA and the SpyCatcher tag after the C-terminal sequence. XynA was cyclized by the formation of an amide bond between Lys in the SpyTag and Asp in Spy Catcher, and the cyclized protein was labeled XynA-Spy.

SDS-PAGE was used to verify the construction of the cyclized enzymes (Figure 10c). There was a clear band at 32.00 kDa, and its enzyme activity was 226.00 U/mL. The molecular weight differed from the theoretical molecular weight of 36.50 kDa, probably because the cyclized protein had a tighter structure and unfolded less under denaturing conditions, which reduced binding with SDS and resulted in faster migration in SDS-PAGE [76]. 

#### 3.3.2. Characterization of Cyclization Mutants

The effects of temperature and pH on the activities of XynA and XynA-Spy are shown in Figure 11. XynA-Spy had the same optimum pH of 5.5 as XynA, and the residual activity of the cyclized mutant remained above 81.78% in the pH range of 4.00–10.00, which was similar to XynA. Zhou used the same method to improve the stability of GH10 family xylanase LXY and found that the mutant with the exogenous peptide had the same optimum pH and pH stability as the original enzyme [38]. In the present study, the optimum temperature of XynA-Spy was 60 °C, which was the same as that of XynA. After being kept at 60 °C for 30 min, the residual activity of XynA-Spy was 62.22%, which was lower than that of XynA (81.51%). After incubation at 80 °C for 30 min, both XynA-Spy and XynA were inactivated, and the half-life of XynA-Spy at 60 °C was 46.83 min, which was 45.95% lower than that of XynA (86.64 min). These results indicated that cyclization did not improve the thermostability of XynA. Waldhauer et al. added the cyclization vector RGKCWE to the C-terminal of *B. subtilis*-derived GH11 xylanase XynA, which lowered the enzyme stability and decreased the *T*_m_ values. However, the addition of only one serine cyclization mutant increased the *T*_m_ value of the enzyme by 7 °C [74]. These results indicate that the selection of a suitable heterologous peptide carrier for the cyclization of protein molecules is important for enhancing of enzyme thermostability [77]. The selection of a suitable cyclization carrier using available information needs to be explored.

### 3.4. Comparison of the Three Strategies for Thermostability Modification

The optimum pH, pH stability, optimum temperature, temperature stability, half-life, and *T*_m_ for XynA and all mutants are summarized in Table 1. Compared with XynA, all mutants showed a same or decrease in pH, which was 5.5 or 5.0, with substitution of different amino acids. All mutations were stable over a wide pH range, except for XynA-SN, which had reduced pH stability under alkaline conditions. The surface entropy predicted that Ala mutation would give a 5 °C increase in optimum temperature and improved thermostability. The largest improvement was observed for disulfide bonds. Three mutants with disulfide bonds had higher optimum temperatures than XynA by up to 10 °C, and all maintained more than 92.80% residual activity when kept at 65 °C for 30 min. The residual activities of the other mutant enzymes dropped below 51.10%. A comparison of the effects of disulfide bonds at various positions on thermostability showed that the disulfide bond at the N-terminal in XynA-VT showed the best thermostability with a half-life of 92.42 min at 70 °C and the highest *T*_m_ of 83.50 °C. This was followed by the mutant enzyme XynA-NT (*T*_m_ of 78.5 °C) with a disulfide bond at the C-terminal. The molecular cyclization technique using the SpyRing exogenous peptide did not improve the thermostability of XynA. The optimum pH, optimum temperature, and temperature stability were similar to those of XynA, and the *T*_m_ was 72 °C, which was lower than that of XynA (73.5 °C). In summary, although single point mutations could improve the thermostability of XynA by changing the interaction forces in local regions, they had a limited effect on the thermostability. Li et al. performed 11 fixed-point mutations and two regional (1–37, 179–188) substitutions in four highly flexible loops of *Aspergillus niger*-derived GH11 xylanase XynA. Iterative saturation mutations, amino acid substitutions by sequence matching, and N-glycosylation were used to construct a combined mutant enzyme xynAm1. Compared with XynA, the half-life of XynAm1 increased by 137.60-fold at 50 °C and reached 156.80 min at 80 °C and 65.40 min at 90 °C. These values are the highest reported for GH11 xylanase to date [78]. The construction of disulfide bonds by mutation, as opposed to multiple amino acid targeted mutations, was a simple and effective way to greatly enhance the thermostability of GH11 xylanase. Xiong et al. constructed a disulfide bond, T2C–T28C, between the N-terminal and β-sheet of *Trichoderma reesei*-derived GH11 xylanase TRX II. They found that the half-life increased from 30 s to 30 min [79]. In the present study, the introduction of only two amino acid mutations to give XynA-VT increased the half-life from 86.64 to 1155.25 min. These results indicated that the construction of disulfide bonds was important to stabilize the secondary structure of the enzyme. The residual activity of XynA-Spy was similar to that of XynA when kept at 65 °C for 30 min. The *T*_m_ of XynA-Spy was lower than XynA, which indicated that the molecular cyclization technique with the SpyRing exogenous peptide did not improve the thermostability of XynA. The introduction of an unsuitable exogenous peptide will influence the normal folding of the protein, and this will thus affect its catalytic function and inhibit enzyme activity.

The molecular dynamics simulation indicated that the N-terminal region was more susceptible to spatial structure disruption under thermal treatment than the catalytic region, and this triggered the thermal unfolding of GH11 xylanases [59,80]. To corroborate the change in structural stability of XynA after mutation, a molecular dynamics simulation was used at 70 °C to compare the RMSD and RMSF of wild-type XynA, K104A, and XynA-VT [81]. The RMSD of XynA was higher than those for K104A and XynA-VT, which showed that reducing the surface entropy and constructing disulfide bonds had positive effects on reducing the stability of the overall conformation. The RMSF values for the N-terminal mutants K104A and XynA-VT were lower than that of XynA, which suggested that the N-terminal of the enzyme was important for stability (Figure 12a,b). The red and green blocks in the N-terminal region (dashed box) for the K104A and XynA-VT were reduced compared with XynA, which indicated that the dynamic correlation of the N-terminal amino acid was reduced (Figure 12c–e). This would have implications for increasing N-terminal stability. Additionally, the RMSF of XynA-VT located near the 125 site was higher than those of K104A and XynA; DCCM showed an antagonistic dynamic correlation between the N-terminal region and site 125 in XynA-VT (dashed box). Although XynA-VT improved stability, it led to fluctuations near site 125 because of the existence of an opposite dynamic correlation, resulting in a higher RMSF value at site 125. Comparison of the N-terminal DCCM of K104A and XynA-VT showed that the red block of XynA-VT decreased, which indicated that the negative correlation between the N-terminal amino acids was weakened and more favorable to the stabilization of the N-terminal as seen in the RMSF. The RMSF of XynA-VT at the N-terminal was lower than that of K104A. Meanwhile, the negative correlation between the N-terminal and the amino acids near site 125 resulted in higher RMSF values near site 125 of XynA-VT than K104A. 

### 3.5. Catalytic and Hydrolytic Properties of XynA and the Mutants

The activities of three mutant enzymes, Q24A, K104A, and K143A, that underwent alanine-targeted mutagenesis after surface entropy prediction of high entropy amino acids were examined with different sources of xylans. The results indicated that XynA and its mutants Q24A, K104A, and K143A had similar substrate preferences. The highest specific enzymatic activity was observed against beechwood xylan, followed by birchwood xylan and oat-spelt xylan. Q24A, K104A, and K143A all showed improved catalytic efficiency compared with XynA for hydrolysis of the different types of xylans. The specific activities for all types of xylan were more than 1.20 times that of XynA (Table 2). The mutants hydrolyzed the different substrates to produce XOS in various yields. When beechwood xylan was used as the substrate, the yield of XOS with Q24A was 12.91% higher than that obtained with XynA. When birchwood xylan was used as the substrate, the hydrolysis yields of mutants Q24A and K104A were 7.76% and 6.02% higher than that obtained with XynA, respectively. When oat-spelt xylan was used as the substrate, the hydrolysis yields of mutants Q24A and K104A were 18.48% and 17.19% higher than that obtained with XynA, respectively. The composition of the hydrolysis products obtained with XynA and the mutants Q24A, K104A, K143A varied slightly. When beechwood xylan and birchwood xylan were used as the substrates, the main products were X2 and X3. When oat-spelt xylan was used as the substrate, the main products were X5, X3, and X2, which accounted for more than 80.00% of the total products. The total amounts of X2, X3, X4, and X5 produced by Q24A within 8 h of hydrolysis for beechwood, birchwood, and oat-spelt xylan were all higher than those produced with XynA (Figure 13). These results showed that Q24A had better catalytic activity than XynA.

The kinetic parameters of the enzyme with beechwood xylan as the substrate are shown in Table 3. Compared with XynA (19.18 mg/mL), the substrate affinity (*K*_m_) of Q24A decreased by 2.30-fold and the catalytic efficiency (*k*_cat_/*K*_m_) increased by 2.06-fold. The *K*_m_ of K143A was 11.75 mg/mL, which was only 61.26% that of XynA, and the catalytic efficiency increased by 1.87-fold compared with XynA. The *K*_m_ of K104A did not change significantly, but its conversion rate *k*_cat_ increased by 1.49-fold and its catalytic efficiency increased by 1.47-fold compared with XynA. The three predicted high-entropy amino acids are located far from the catalytic cleavage site of XynA, so they will not be directly involved in the enzyme’s catalytic function [46]. Xing et al. combined error-prone PCR and B-factor analysis to improve the factors influencing the thermostability of GH11 family xylanases and found that most of the beneficial mutations occurred on the enzyme surface away from the catalytic active center, such as in the loop region [68]. Additionally, the GRAVY values of all three mutant enzymes were reduced compared with XynA, which indicated that the hydrophobicity was enhanced (Figure 6). Yu et al. confirmed that amino acid alterations away from the active center indirectly affected enzyme–substrate interactions at the catalytic cleft, which affected enzyme–substrate binding and catalytic efficiency [81,82]. In summary, reducing surface entropy and enhancing hydrophobicity at the active site positions could improve the catalytic efficiency of XynA by affecting the affinity and conversion of the enzyme to the substrate, which will improve the thermostability.

The construction of disulfide bonds within enzymes also stabilizes the enzyme structure by reducing the conformational entropy [23]. However, although disulfide bonds have an important role in improving the thermostability of enzymes, they also often affect the catalytic efficiency [83] and the hydrolytic properties of enzymes [48,52]. The positioning of disulfide bonds should be selected to avoid adverse effects on the correct folding of the enzyme and the redox environment. In this study, disulfide bonds were introduced in the more flexible regions of the protein, at the N-, and C-terminals, and in the region between the α-helix and β-sheet B9, and the effects on the catalytic activity and hydrolysis product properties were investigated. The substrate specificities of the enzymes for different types of xylans were examined, and the specific enzyme activities of the three recombinant enzymes are shown in Table 2. All xylanases showed the highest specific activity for beechwood xylan and the lowest specific activity for oat-spelt xylan. When beechwood xylan was used as the substrate, the specific activities of XynA-VT, XynA-NT, and XynA-SN were 1963.81, 2017.74, and 2010.55 U/mg, respectively. These values for XynA-VT, XynA-NT, and XynA-SN were 44.51%, 48.49%, and 47.96% higher than that of XynA (1358.86 U/mg), respectively. The construction of a disulfide bond improved the thermostability of xylanase and also enhanced its specific activity. The composition and proportion of hydrolysis products for XynA, XynA-VT, XynA-NT, and XynA-SN were similar when they hydrolyzed different types of xylan. When beechwood and birchwood xylans were used as substrates, the main products were X2 and X3, which accounted for approximately 90% of the total products. When oat-spelt xylans were used as the substrates, the main products were X5, X3, and X2, which accounted for approximately 90% of the total products. In terms of product yield, when the mutants formed by constructing disulfide bonds were used to hydrolyze the three different sources of xylan, higher yields were obtained for XOS than for XynA. The highest yield was obtained with XynA-VT in hydrolysis of birchwood xylan for 8 h. The total amount of XOS was 1.35-fold higher than that obtained with XynA, and more X4 and X5 were produced. The kinetic parameters of the mutants with a disulfide bond are shown in Table 3. Compared with XynA, all three mutants had improved catalytic efficiencies and the *k*_cat_/*K*_m_ increased by more than 1.50-fold. The *K*_m_ of XynA-VT and XynA-SN were not significantly different from that of XynA, but their *k*_cat_ were higher, at 2265.06 and 1803.67 /s, respectively. The corresponding *k*_cat_/*K*_m_ values were 79.91% and 60.68%, respectively.

Molecular docking of XynA and the mutant enzymes with substrate X6 showed that the binding energies of K104A and XynA-VT with X6 were −3.82 and −4.31 kJ/mol, respectively. These values were higher than that of XynA (−4.58 kJ/mol). The binding energies of Q24A, K143A, XynA-NT, and XynA-SN were −6.58, −5.04, −5.59, and −5.05 kJ/mol, respectively, which were all lower than that of XynA. The lower binding energies indicated higher affinity to the substrate. These results were consistent with the determination of kinetic parameters, and the *K*_m_ of both K104A and XynA-VT were slightly higher than that of XynA. The improvement in the catalytic efficiency was associated with a higher conversion rate, binding energy, and hydrogen bonding interactions, which led to the increased catalytic efficiency of the mutants.

XynA-VT contained a disulfide bond in the N-terminal region of the enzyme structure, which was the initiation region of xylanase unfolding and located far from the catalytic activity center of the enzyme. The more stable N-terminal of XynA-VT improved the thermostability [59]. Studies have shown that the construction of disulfide bonds between N-terminal amino acids and β-sheet A2 could enhance the enzyme rigidity while indirectly affecting the binding of the enzyme and the substrate. This will be conducive to achieve both thermostability and catalytic activity of the enzyme [24,52,84]. The construction of disulfide bonds at the N-terminal had positive effects on the thermostability, catalytic activity, and hydrolytic properties of GH11 xylanase.

The substrate specificity of XynA-Spy was measured under the optimum conditions using 2% beechwood, birchwood, and oat-spelt xylan as substrates. The cyclized mutant enzyme showed similar hydrolytic activity to XynA, and the activity of XynA-Spy decreased in the order of beechwood, birchwood, and oat-spelt xylan (Table 2). If the activity of the beechwood xylan substrate was defined as 100.00%, the activities of XynA-Spy for birchwood xylan and oat-spelt xylan were 51.63% and 34.81%, respectively. The corresponding activities of XynA were 46.39% and 35.33%. The cyclization did not change the substrate specificity of the enzyme. Using beechwood and birchwood xylan as substrates, the main products were X1, X2, and X3. The proportion of X3 decreased and the proportion of X1 increased after 8 h of hydrolysis compared with XynA. Using oat-spelt xylan as the substrate, the main products were X2, X3, and X5. The introduction of heterologous peptides both reduced the hydrolysis efficiency of XynA, and changed the composition of the hydrolytic products, which reduced production of X3 and increased production of X1. Compared with XynA, the substrate affinity of XynA-Spy decreased. Its *K*_m_ was 21.57 mg/mL, which was slightly higher than that of XynA (19.18 mg/mL). The catalytic efficiency of XynA-Spy was 41.07 mL/s/mg, which was about two-thirds of that of XynA (62.98 mL/s/mg).

The recombinant enzyme XynA-Spy, which was constructed by molecular cyclization of the fused heterologous peptide, showed decreases in thermostability and catalytic activity compared with XynA. The introduction of SpyTag and SpyCatcher did not improve the thermostability of XynA, even though the enzyme terminals were immobilized. It was presumed that the length of the heterologous peptide was too long, leading to a change in the conformation of the enzyme and inhibition of the enzyme–substrate binding. The introduction of SpyTag and SpyCatcher also decreased the affinity, hydrolysis efficiency, and production rate of xylanase toward substrates, which decreased the production of X3 and increased the production of X1. It was assumed that the exogenous peptide hindered the binding of the enzyme to the substrate, and the release of the product affected the mode of action of the enzyme with the substrate. Although the molecular cyclization did improve the enzyme properties, the terminal of GH11 xylanase played an important role in the change in enzyme conformation, which influenced the enzymatic stability, and catalytic and hydrolysis properties.

## 4. Conclusions

In this study, three strategies were used to modify XynA and stabilize its conformation. Enzymatic property characterization and structure simulation analysis showed that the thermostability, hydrolytic properties, and catalytic abilities of XynA were greatly improved by reduction of the surface entropy or the construction of disulfide bonds. A disulfide bond at the N-terminal gave the most significant effect, and this modification produced a good candidate, XynA-VT, for XOS production. Molecular cyclization incorporating SpyRing exogenous peptide did not improve the thermostability or catalytic activity of XynA, which may be related to the selection of suitable cyclization carriers. The three modification strategies provide feasible options for the rational design of GH11 xylanase with improved thermostability and catalytic properties, and a reference for the engineering of related enzymes.

## Figures and Tables

**Figure 1 foods-12-00879-f001:**
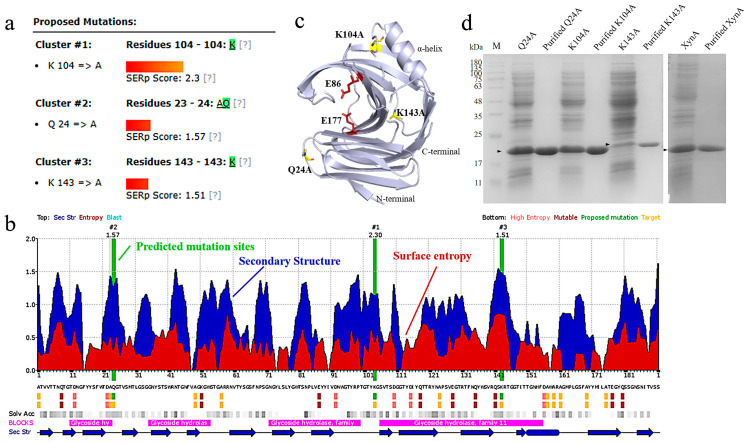
Prediction of surface entropy mutations and purification of the mutants. (**a**) The total clustering score (SERp score) of the predicted mutant loci indicates the success rate of the prediction; (**b**) Results of the overall analysis of surface entropy distribution, secondary structure prediction, and sequence conservativeness (the green vertical bars indicated the predicted mutation sites; the blue peaks indicated the loop area of secondary structure prediction; the red peaks indicated the predicted sites with high surface entropy; the bottom color block was red for variable residues; pink for high entropy residues; green for suggested mutation residues; yellow for low entropy target Solv Acc; gray marked part was the solvent accessibility of the predicted amino acid residues; and the darker color indicated higher solvent accessibility; BLOCKS purple marked part was the highly conserved region; Sec Str blue marked part was the predicted secondary structure). (**c**) 3D structural model of xylanase XynA; the three mutant sites were marked yellow; the active sites were marked red. (**d**) SDS-PAGE analysis of XynA and its mutants before and after purification. Lane M, protein marker.

**Figure 2 foods-12-00879-f002:**
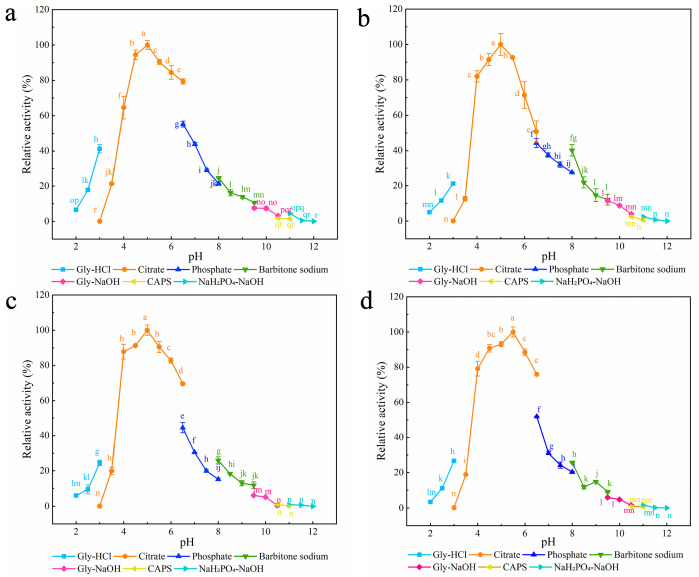
The optimum pH of SER mutated enzyme and XynA. (**a**) Q24A; (**b**) K104A; (**c**) K143A; (**d**) XynA. The optimal pH was measured at 37 °C for 10 min in different buffers (50 mM) from 2.0 to 12.0, using beechwood xylan as substrate; buffers used were 50 mM Gly-HCl buffer (2.0–3.0), citrate buffer (pH 3.0–6.5), phosphate buffer (pH 6.5–8.0), barbital sodium buffer (pH 8.0–9.5), Gly-NaOH buffer (pH 9.5–10.5), CAPS buffer (pH 10.5–11.0), NaH_2_PO_4_-NaOH buffer (pH 11.0–12.0); the highest enzyme activity was used as 100%. Means within rows followed by the same letter were not significantly different (*p* < 0.05).

**Figure 3 foods-12-00879-f003:**
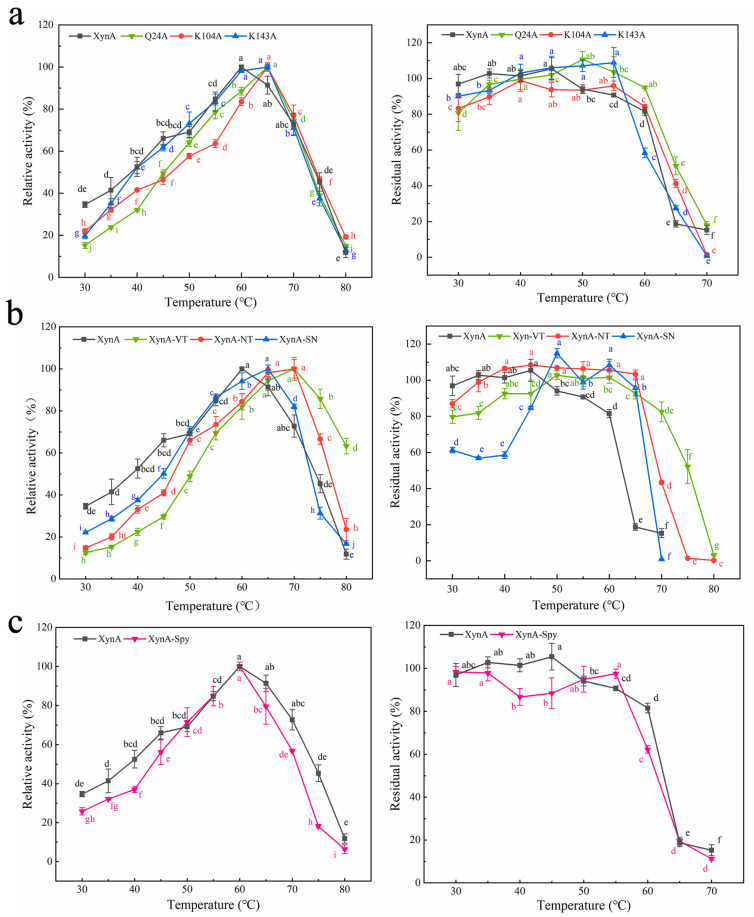
The optimal temperature (**left**) and thermostability (**right**) of XynA and its mutants. The optimum temperature was measured in 50 mM citrate buffer (optimum pH) at different temperatures (30–80 °C), and the highest enzyme activity was used as 100%; thermostability was determined by incubating at varying temperatures (30–80 °C) in 50 mM citrate buffer (optimum pH) for 30 min; the activity of untreated xylanase was defined as 100%. (**a**) The optimum temperature and the thermostability of SER mutated enzyme. (**b**) The optimum temperature and the thermostability of disulfide mutant enzyme. (**c**) The optimum temperature and the thermal stability of XynA-Spy. Means within rows followed by the same letter were not significantly different (*p* < 0.05).

**Figure 4 foods-12-00879-f004:**
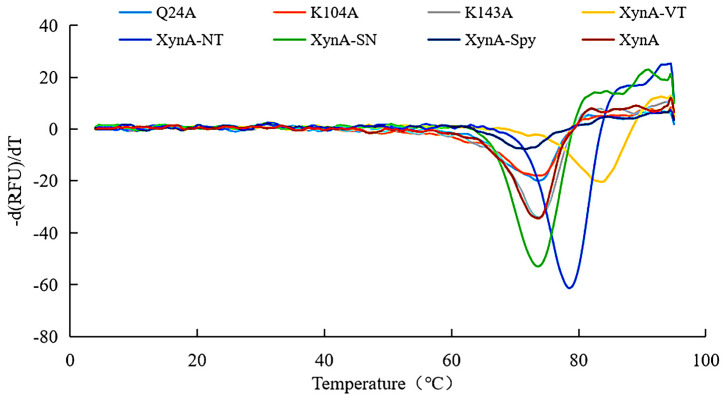
The *T*_m_ of XynA and its mutants.

**Figure 5 foods-12-00879-f005:**
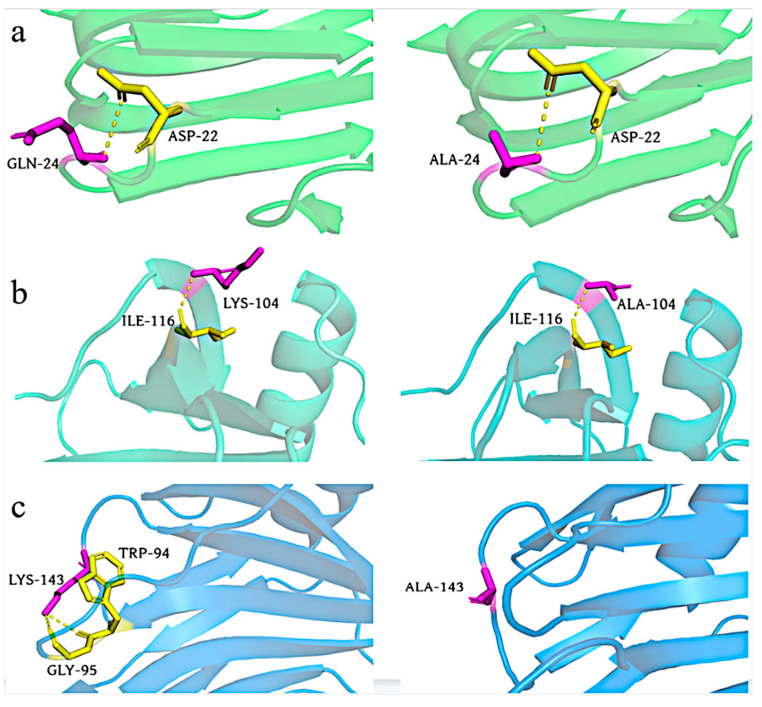
Model structures of the wild type (**left**) and the XynA mutant (**right**); yellow dashed lines represent hydrogen bonds. Novel hydrogen bonding interaction discovered in three single mutants, and illustration of the newly formed hydrogen bonding interaction and analogous location in Q24A (**a**), K104A (**b**), and K143A (**c**).

**Figure 6 foods-12-00879-f006:**
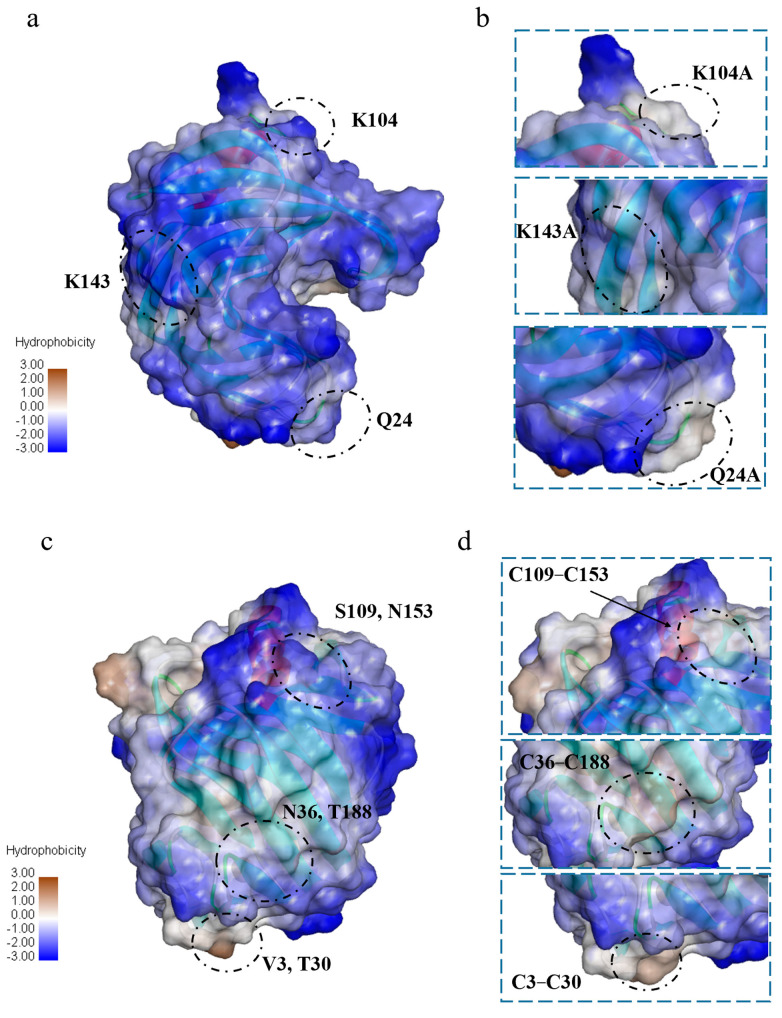
Analysis of hydrophilic and hydrophobic results of SER mutated enzyme (**b**), disulfide bond mutated enzyme (**d**), and XynA (**a**,**c**). Blue for hydrophilic, and orange for hydrophobic.

**Figure 7 foods-12-00879-f007:**
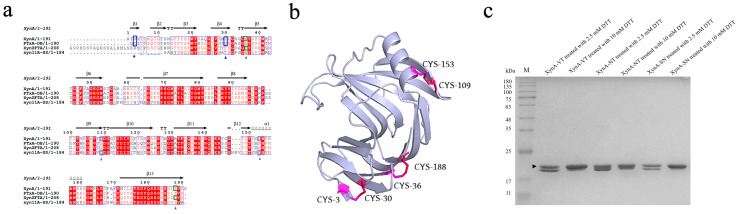
Disulfide bond construction position and expression purification of its disulfide bond-containing mutants. (**a**) Multiple amino acid sequence alignment of the XynA with other family GH11 thermostable xylanases (the mutation sites were marked in different colors, and the pair of amino acids that constructed the disulfide bond were in the same color). (**b**) Position of disulfide bond construction on the 3D model structure of XynA; the mutant sites were marked in pink and red. (**c**) SDS-PAGE analysis of xylanase and validation of the disulfide bonds formation. Lane M, protein marker.

**Figure 8 foods-12-00879-f008:**
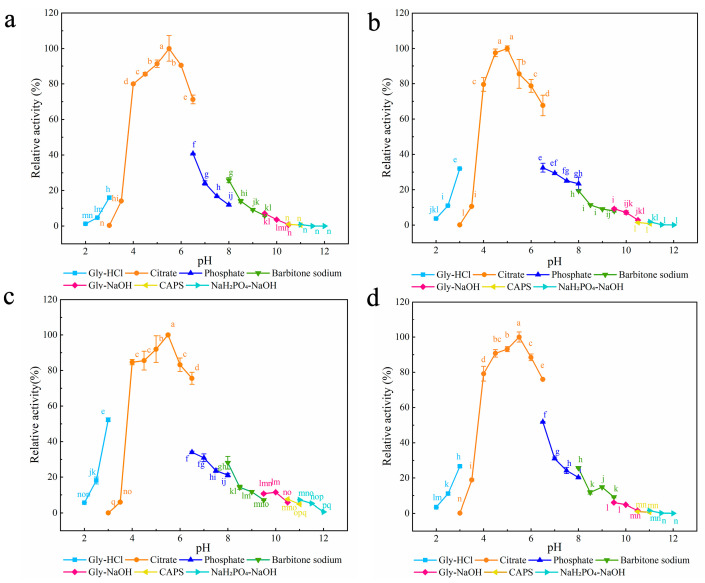
The optimum pH of disulfide mutant enzymes and XynA. (**a**) XynA-VT; (**b**) XynA-NT; (**c**) XynA-SN; (**d**) XynA. The optimal pH was measured at 37 °C for 10 min in different buffers (50 mM) from 2.0 to 12.0, using beechwood xylan as substrate; buffers used were 50.00 mM Gly-HCl buffer (2.0–3.0), citrate buffer (pH 3.0–6.5), phosphate buffer (pH 6.5–8.0), barbital sodium buffer (pH 8.0–9.5), Gly-NaOH buffer (pH 9.5–10.5), CAPS buffer (pH 10.5–11.0), NaH_2_PO_4_-NaOH buffer (pH 11.0–12.0); the highest enzyme activity was used as 100%. Means within rows followed by the same letter were not significantly different (*p* < 0.05).

**Figure 9 foods-12-00879-f009:**
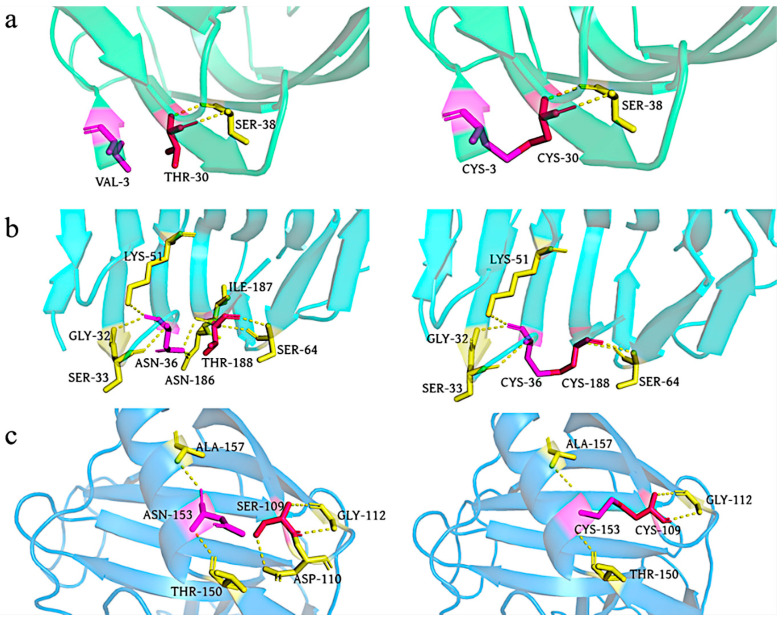
Model structures of the wild type (**left**) and the XynA mutant (**right**). Yellow dashed lines represent hydrogen bonds. Novel hydrogen bonding interaction discovered in three single mutants, illustration of the newly formed hydrogen bonding interaction and analogous location in XynA-VT (**a**), XynA-NT (**b**), and XynA-SN (**c**).

**Figure 10 foods-12-00879-f010:**
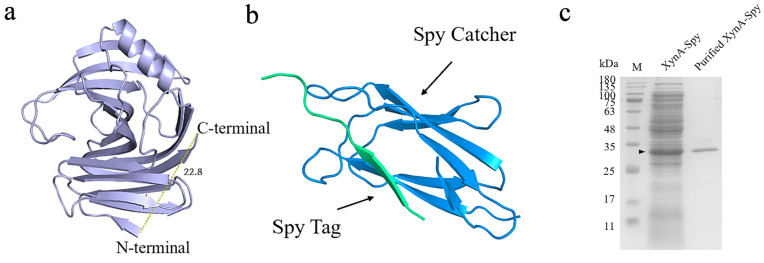
The 3D model structure of XynA and SpyRing. (**a**) The 3D model structure of XynA; the yellow dashed line connects the N-terminal with the C-terminal, and the numbers showed their distances. (**b**) The 3D model structure of SpyRing was composed of two parts, Spy Tag and Spy Catcher. (**c**) SDS-PAGE analysis of XynA and XynA-Spy before and after purification. Lane M, protein marker.

**Figure 11 foods-12-00879-f011:**
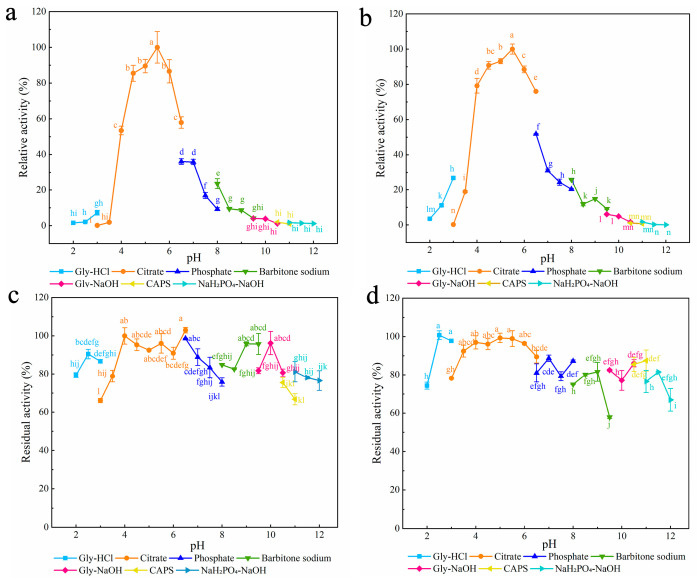
The optimum pH and pH stability of XynA-Spy and XynA. The optimal pH was measured at 37 °C for 10 min in different buffers (50 mM) from 2.0 to 12.0, and the highest enzyme activity was used as 100%; pH stability was determined by incubating at varying pH from 2.0 to 12.0 at 37 °C for 30 min, and the activity of untreated xylanase was defined as 100%. Using beechwood xylan as substrate, buffers used were 50 mM Gly-HCl buffer (2.0–3.0), citrate buffer (pH 3.0–6.5), phosphate buffer (pH 6.5–8.0), barbital sodium buffer (pH 8.0–9.5), Gly-NaOH buffer (pH 9.5–10.5), CAPS buffer (pH 10.5–11.0), NaH_2_PO_4_-NaOH buffer (pH 11.0–12.0). (**a**) The optimum pH of XynA-Spy. (**b**) The optimum pH of XynA. (**c**) The pH stability of XynA-Spy. (**d**) The pH stability of XynA. Means within rows followed by the same letter were not significantly different (*p* < 0.05).

**Figure 12 foods-12-00879-f012:**
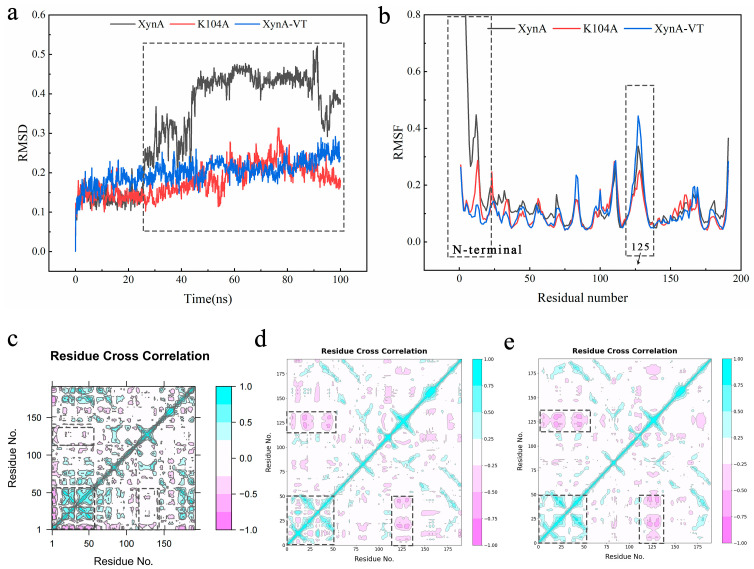
The molecular dynamics simulation of enzymes. (**a**) RMSD of XynA, K104A, and XynA-VT121 were analyzed over the whole simulation at 70 °C. (**b**) RMSF of XynA, K104A, and XynA-VT121 were analyzed over the whole simulation at 70 °C. (**c**) Dynamical cross-correlation matrix of XynA; green indicates positive correlation, and red indicates negative correlation. (**d**) Dynamical cross-correlation matrix of K104A. (**e**) Dynamical cross-correlation matrix of XynA-VT.

**Figure 13 foods-12-00879-f013:**
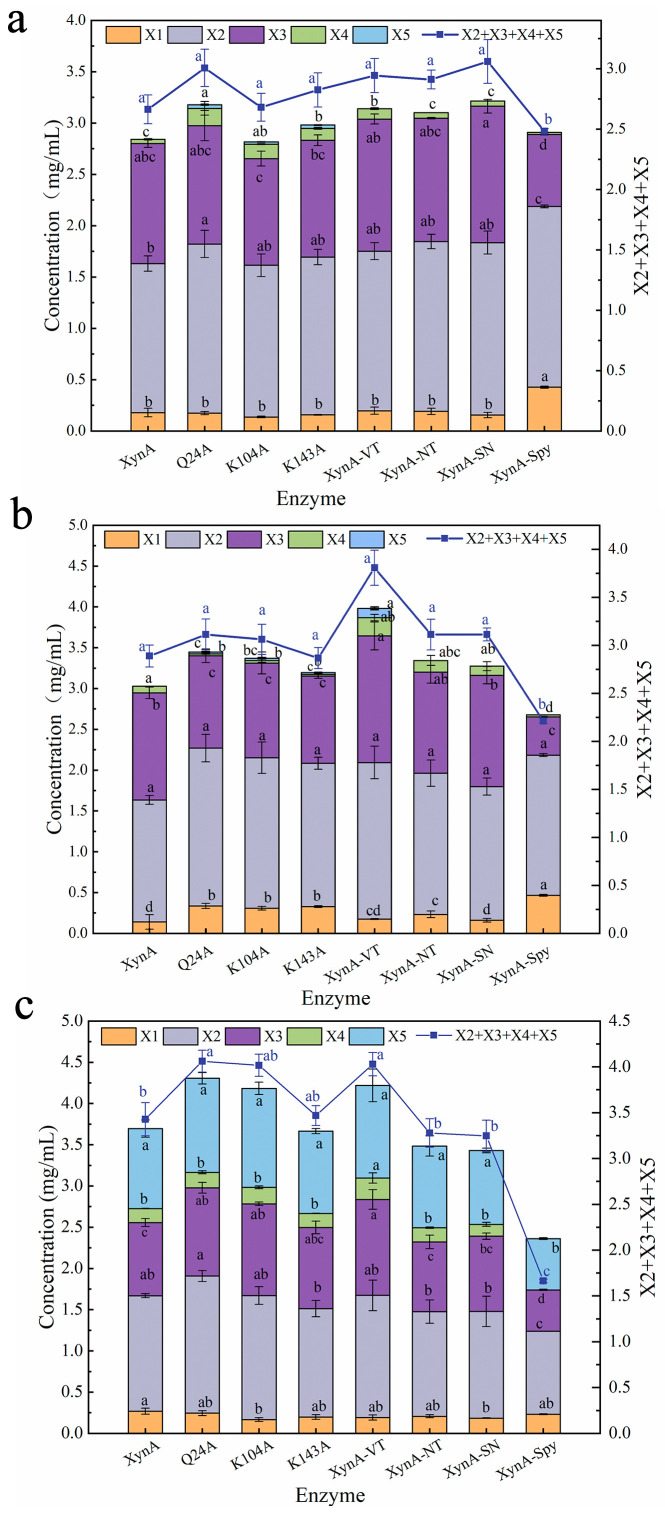
The yield of hydrolytic products of wild-type XynA and mutants with beechwood xylan (**a**), birchwood xylan (**b**), and oat-spelt xylan (**c**) at pH 5.0, 50 °C for 8 h. XOS standard markers, X1, xylose; X2, xylobiose; X3, xylotriose; X4, xylotetraose; and X5, xylopentaose. Means within rows followed by the same letter were not significantly different (*p* < 0.05).

**Table 1 foods-12-00879-t001:** Characterization of XynA and its mutants.

	XynA	Q24A	K104A	K143A	XynA-VT	XynA-NT	XynA-SN	XynA-Spy
Optimum pH	5.5 ^a^	5.0 ^b^	5.0 ^b^	5.0 ^b^	5.5 ^a^	5.0 ^b^	5.5 ^a^	5.0 ^b^
pH stability rage (>80%)	3.5–9.0 ^h^	2.0–8.5 ^e^	2.0–11.5 ^a^	2.0–11.0 ^b^	2.0–10.0 ^c^	2.0–9.5 ^d^	2.5–7.5 ^g^	2.5–10.5 ^f^
Optimum temperature (°C)	60 ^d^	65 ^b^	65 ^b^	60–65 ^c^	70 ^a^	70 ^a^	65 ^b^	60 ^d^
Thermostability (65 °C, 30 min)	20% ^d^	>40% ^b^	>40% ^b^	>20% ^c^	>90% ^a^	>90% ^a^	>90% ^a^	20% ^d^
*t*_1/2_^60°C^ (min)	86.64 ^d^	76.17 ^e^	88.87 ^c^	69.31 ^f^	1155.25 ^a^	693.15 ^b^	nd ^h^	46.83 ^g^
*t*_1/2_^70°C^ (min)	nd ^d^	nd ^d^	nd ^d^	nd ^d^	92.42 ^a^	35.01 ^b^	210.04(s)^c^	nd ^d^
*T*_m_ (°C)	73.5 ^c^	73.5 ^c^	73.5 ^c^	73.5 ^c^	83.5 ^a^	78.5 ^b^	73.5 ^c^	72 ^d^

Means within columns followed by the same letter were not significantly different (*p* < 0.05); nd: not detectable.

**Table 2 foods-12-00879-t002:** Specific enzyme activity of polymeric substrates of XynA and its mutants.

	XynA	Q24A	K104A	K143A	XynA-VT	XynA-NT	XynA-SN	XynA-Spy
Beechwood xylan (U/mg)	1358.86 ± 151.42 ^a^	1710.65 ± 58.88 ^a^	1532.84 ± 55.30 ^a^	1710.01 ± 59.64 ^a^	1963.81 ± 11.65 ^a^	2017.74 ± 106.78 ^a^	2010.55 ± 151.42 ^a^	400.38 ± 54.05 ^a^
Birchwood xylan (U/mg)	630.40 ± 37.70 ^b^	920.98 ± 2.26 ^b^	1058.89 ± 24.84 ^b^	1069.93 ± 33.26 ^b^	1585.67 ± 9.22 ^b^	1147.59 ± 96.21 ^b^	1198.67 ± 212.03 ^b^	206.72 ± 39.85 ^b^
Oat-spelt xylan (U/mg)	480.10 ± 29.30 ^c^	806.64 ± 26.30 ^c^	1013.57 ± 102.80 ^b^	1013.16 ± 5.73 ^b^	923.56 ± 45.62 ^c^	432.98 ± 64.11 ^c^	401.85 ± 105.75 ^c^	139.39 ± 11.79 ^c^

Values are the mean of three replicate. Means within columns followed by the same letter were not significantly different (*p* < 0.05).

**Table 3 foods-12-00879-t003:** Kinetic parameters of XynA and the mutants on beechwood xylan.

	XynA	Q24A	K104A	K143A	XynA-VT	XynA-NT	XynA-SN	XynA-Spy
*K*_m_ (mg/mL)	19.18 ± 1.67 ^b^	8.31 ± 0.10 ^d^	19.46 ± 0.70 ^ab^	11.75 ± 1.02 ^c^	19.99 ± 1.24 ^ab^	13.26 ± 0.97 ^c^	17.83 ± 0.70 ^b^	21.57 ± 2.02 ^a^
*V*_max_ (μmol/min/mg)	3484.00 ± 165.00 ^c^	3120.00 ± 516.00 ^c^	5185.00 ± 577.00 ^b^	3993.00 ± 135.00 ^c^	6524.43 ± 215.98 ^a^	3775.05 ± 487.55 ^c^	5195.38 ± 1237.22 ^b^	1417.45 ± 377.17 ^d^
*k*_cat_ (/s)	1208.00 ± 57.00 ^cd^	1080.00 ± 179.00 ^cd^	1795.00 ± 200.00 ^b^	1382.00 ± 47.00 ^c^	2265.06 ± 74.75 ^a^	1309.94 ± 168.76 ^c^	1803.67 ± 428.25 ^b^	885.67 ± 130.55 ^d^
*k*_cat_/*K*_m_ (mL/s/mg)	62.98 ± 7.78 ^e^	129.96 ± 12.83 ^a^	92.24 ± 3.12 ^d^	117.62 ± 3.08 ^ab^	113.31 ± 1.14 ^bc^	98.72 ± 8.52 ^d^	101.16 ± 9.29 ^cd^	41.07 ± 9.05 ^f^

Values are the mean of three replicate. Means within columns followed by the same letter were not significantly different (*p* < 0.05).

## Data Availability

Data are contained within the article. All the data generated for this study are available on request to the corresponding author.

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
