# Peer review of "Three Molecular Modification Strategies to Improve the Thermostability of Xylanase XynA from Streptomyces rameus L2001"

_foods, 2023, doi:10.3390/foods12040879_

Round 1

Reviewer 1 Report

Dear Editors and authors, 

The manuscript (Three molecular modification strategies to improve the thermostability of xylanase XynA from Streptomyces rameus L2001) has good idea but it needs some corrects and modifications. 

1-Gene names should be written in italics throughout the manuscript. See Page 1 line 17 ,......... etc. 

2-What is the source of E. coli DH5α bacteria?

3-Many of the work methods in the manuscript do not contain scientific references. See Expression and purification of xylanases, Biochemical characterization of xylanase,  Quantitative real-time PCR and standard assay procedure.

4-The authors did not mention anything about the statistical analysis in the work methods chapter, but we note that Table 2, 3 and figure 13 contains a statistical analysis.

5-Some of the results in the results chapter need a statistical analysis in order to know the amount of significant differences between the treatments. See figure 2, 3, 6, 9 and table 1.

6- Figure 1d, figure 5c, and figure 8 c,  What is the size of the bunds that appeared in the results? Silent pictures are not sufficient to illustrate the results.

7-The conclusions contain some results that should be removed.

Reviewer 2 Report

The paper presents some strategies for improving thermostability of xylanase XynA from Streptomyces rameus. The manuscript is well structured and very complex analyses were presented. Some remarks are made below. 

Abstract: L18, 24 Please see English. Sentences need revision. 

The introduction provides enough background for the study.

Materials and methods: Please explain abbreviations.

Please add the number of replicates for each determination. 

Statistics are missing.

Results and discussion: L311 This information should be moved to methods. 

L322. English error, please check.

L338. Please move to methods. 

Table 1: statistics and standard deviations are missing. ANOVA and post hoc tests can be applied to evaluate the differences among samples.

Figure 11 must be mentioned before in the text.  

Table 2, 3: see suggestions for Table 1.

I suggest increasing the graphics from figure 13. 

Conclusion: Please include the limitations of the study and further perspectives. 

Round 2

Reviewer 2 Report

The paper was improved. I recommend increasing the size of the figures because it is difficult to distinguish the letters for statistics.